# Modeling of Perforated Piezoelectric Plates

**Houari Mechkour**

ECE-Paris Engineering School, 37 Quai de Grenelle, CS-71520, CEDEX 15, 75015 Paris, France; mechkour@ece.fr

**Abstract:** In this article, we are interested in the behavior of a three-dimensional model of periodic perforated piezoelectric plate, when the thickness $h$ of the plate and the size $\varepsilon$ of the holes are small. We study the dependence of displacements and electric potential on $h$ and $\varepsilon$, and give equivalent limits when $h$ and $\varepsilon$ tend towards zero. We compute analytical formulae for all effective properties of the periodic perforated piezoelectric plate.

**Keywords:** asymptotic analysis; homogenization; piezoelectricity; plates; perforations

## 1. Introduction

Piezoelectric plates and, more generally, structures containing piezoelectric materials are widely used in engineering applications, such as sensors or actuators. The piezoelectric effect is the capacity exhibited by some materials to convert a mechanical deformation to electric field and vice versa. Application to the materials of an electric field produces a mechanical deformation. The increased application of composite and perforated (lattice) piezoelectric materials in ground-breaking Micro–Electro–Mechanical-Systems (MEMS) has stimulated great interest in several studies (see Ikeda [1]).

In the literature, there are several papers concerning modeling on piezoelectric structures. We refer, in particular, to the works of Rahmoune [2,3] and Sene [4,5], who modeled the behavior of a piezoelectric static thin plate not perforated with the thickness $h$ tending to zero. In [6], the authors modeled the thin piezoelectric shell. Kauffman and Saint Jean-Paulin [7] studied the behavior of the displacement when the parameter of the thickness of the plate, period of perforation, and the ration between the width of the bar and the period tended to zero. Figueiredo and Leal [8] used asymptotic analysis to obtain a 2D piezoelectric model.

The present paper is inspired by Rahmoune [2,3], and Sene [4,5], and again, we use the asymptotic analysis and homogenization theory (see [9] or [10]) to derive a reduced piezoelectric model and we give the homogenized system. Explicit formulae of elastic, piezoelectric, and dielectric homogenized coefficients are reported.

## 2. Setting of the Problem

In this section, we first introduce some notations. Then, we recall the static three-dimensional piezoelectric model of non-homogeneous anisotropic thin plate, and we describe its formulation as a boundary value problem and the variational formulation. Throughout this paper, $L^2(\Omega)$ in the Sobolev space of real-valued functions that are measurable and square summable in $\Omega$ with respect to the Lebesgue measure. We use $C^{\infty}_{\sharp}(Y)$ to denote the space of infinitely differentiable functions in $\mathbb{R}^3$ that are periodic of $Y$. The subscript $\sharp$ stands for $Y$-periodic functions in the last variable.

### 2.1. Geometric of the Medium

Let $\omega_{\varepsilon}$ be a bounded perforated domain of $\mathbb{R}^2$, the boundary of which $\partial\omega_{\varepsilon}$ is regular. The three dimensional piezoelectric domain $\Omega_{h\varepsilon}$ is defined in the following way (see Figure 1)

$$\Omega_{h\varepsilon} = \omega_{\varepsilon} \times ]-h, +h[, \quad h \in \mathbb{R}^*,$$

where $\Omega_{h\varepsilon}$ is plate with middle surface $\omega_\varepsilon$ and thickness $2h$. $T_{h\varepsilon} = t_\varepsilon \times ] - h, +h[$ is the set of the cylindrical perforations. The upper (resp. lower) face is $\Gamma_{h\varepsilon}^+ = \omega_\varepsilon \times \{+h\}$ (resp. $\Gamma_{h\varepsilon}^- = \omega_\varepsilon \times \{-h\}$) and $\Gamma_{h\varepsilon}^l = \gamma \times ] - h, +h[$ is the exterior lateral boundary of the plate. We set $\Gamma_{h\varepsilon} = \partial \Omega_{h\varepsilon}$.

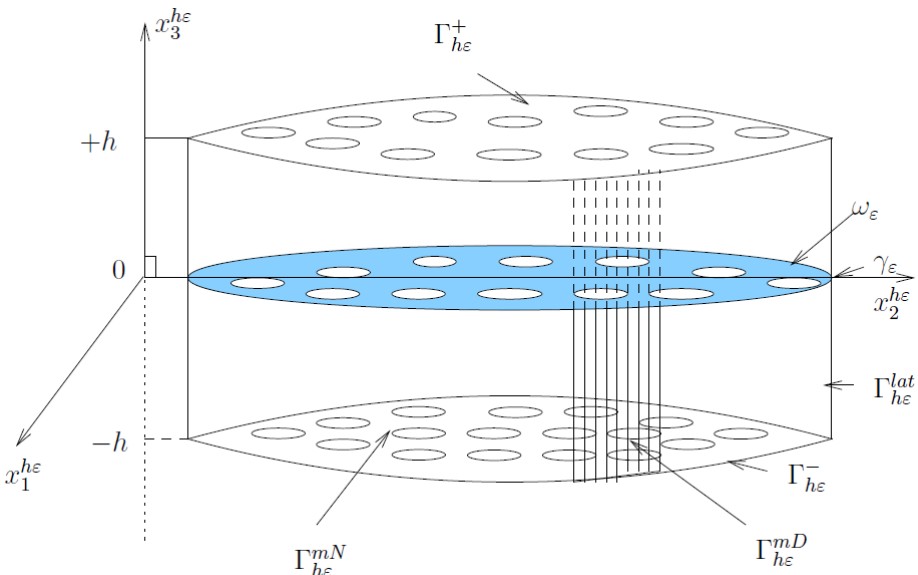

**Figure 1.** Perforated piezoelectric plate.

The plate is clamped on the exterior lateral boundary (the Dirichlet conditions) in the placement in $\Gamma_{h\varepsilon}^{mD} = \gamma_\varepsilon^{m0} \times ] - h, +h[$, with $\mathrm{mes}(\gamma_\varepsilon^{m0}) > 0$. We use $\Gamma_{h\varepsilon}^{mN}$ to denote the complementary portion of the lateral surface. We have

$$\Gamma_{h\varepsilon}^{mN} = \Gamma_{h\varepsilon}^+ \cup \Gamma_{h\varepsilon}^- \cup (\gamma_\varepsilon^{m1} \times ] - h, +h[) = \Gamma_{h\varepsilon} - \Gamma_{h\varepsilon}^{mD}, \quad \text{where } \gamma_\varepsilon^{m0} \cup \gamma_\varepsilon^{m1} = \partial \omega_\varepsilon.$$

We have Neumann boundary conditions on the boundary of the holes on the top and bottom faces.

### 2.2. Model Problem

The unknown of the piezoelectric plate model is the pair $(\mathbf{u}^{h\varepsilon}, \varphi^{h\varepsilon})$, where $\mathbf{u}^{h\varepsilon} = (u_1^{h\varepsilon}, u_2^{h\varepsilon}, u_3^{h\varepsilon})$ denotes the displacement vector field and $\varphi^{h\varepsilon}$ is the electric potential, which is a scalar field. The current point in $\Omega_{h\varepsilon}$ is denoted by $x = (x_1, x_2, x_3)$. The plate under consideration is made of linearly piezoelectric and anistropic body; the elastic, piezoelectric and electric moduli are periodic for the variables $x_1$ and $x_2$. The period is the order of a small-parameter $\varepsilon$.

The equations of equilibrium and Gauss's law of electrostatics, in the absence of free charges, are written as:

$$\begin{cases} -\mathbf{div}\,\boldsymbol{\infty}^{h\varepsilon}(\mathbf{u}^{h\varepsilon}, \phi^{h\varepsilon}) &= \mathbf{f}^h \quad \text{in } \Omega_{h\varepsilon}, \\ -\mathbf{div}\,\mathbf{D}^{h\varepsilon}(\mathbf{u}^{h\varepsilon}, \phi^{h\varepsilon}) &= 0 \quad \text{in } \Omega_{h\varepsilon}, \end{cases} \tag{1}$$

we complete the boundary conditions,

$$\begin{cases} \mathbf{u}^{h\varepsilon} &= \mathbf{0} \quad \text{on } \Gamma^{h\varepsilon}, \\ \phi^{h\varepsilon} &= 0 \quad \text{on } \Gamma^{h\varepsilon}, \\ \sigma^{h\varepsilon}(\mathbf{u}^{h\varepsilon}, \phi^{h\varepsilon}).n^{h\varepsilon} &= 0 \quad \text{on } \partial T_{h\varepsilon} \cup \Gamma_{h\varepsilon}^{\pm}, \\ \mathbf{D}^{h\varepsilon}(\mathbf{u}^{h\varepsilon}, \phi^{h\varepsilon}).n^{h\varepsilon} &= 0 \quad \text{on } \partial T_{h\varepsilon} \cup \Gamma_{h\varepsilon}^{\pm}, \end{cases} \tag{2}$$

where $\mathbf{f}^h \in \mathbf{L}^2(\Omega_{h\varepsilon})$ (in fact, $\mathbf{f}^h$ refers to the restriction of $\mathbf{f}$ in $\Omega_{h\varepsilon}$). The second-order stress tensor $\sigma^{h\varepsilon} = (\sigma_{ij}^{h\varepsilon})$ (In the following, we adopt the Einstein convention, with respect to the summation of repeated indices, and the Latin indices run from 1 to 3, Greek indices (except $\varepsilon$) taking values in $\{1,2\}$) and the electric displacement vector $\mathbf{D}^{h\varepsilon} = (D_i^{h\varepsilon})$ are linearly related to the second-order $s_{kl}(\mathbf{u}) = \frac{1}{2}(\partial_k \mathbf{u}_l + \partial_l \mathbf{u}_k)$ and the scalar electric field $\partial_k \phi^\varepsilon$ by the constitutive law strain tensor

$$
\begin{cases}
\text{œ}_{ij}^{h\varepsilon}(\mathbf{u}^{h\varepsilon}, \phi^{h\varepsilon}) = c_{ijkl}^{h\varepsilon} s_{kl}(\mathbf{u}^{h\varepsilon}) + e_{kij}^{h\varepsilon} \partial_k \phi^{h\varepsilon} & \text{in } \Omega_{h\varepsilon}, \\[2mm]
D_i^{h\varepsilon}(\mathbf{u}^{h\varepsilon}, \phi^{h\varepsilon}) = -e_{ikl}^{h\varepsilon} s_{kl}(\mathbf{u}^{h\varepsilon}) + d_{ij}^{h\varepsilon} \partial_j \phi^{h\varepsilon} & \text{in } \Omega_{h\varepsilon}.
\end{cases}
\tag{3}
$$

$$
1 \le i, j, k, l \le 3
$$

avec $(\mathbf{div}\ \text{œ}^{h\varepsilon})^i = \partial_j \sigma_{ij}^{h\varepsilon}$, $div\ \mathbf{D}^{h\varepsilon} = \partial_i D_i^{h\varepsilon}$, $\partial_i = \frac{\partial}{\partial x_i}$, $x = (x_i) \in \Omega$, The material properties are given by the fourth-order stiffness tensor $c_{ijkl}^{h\varepsilon}$ measured at constant electric field. The elastic coefficients satisfies the symmetries conditions, and ellipticity uniformly in $\varepsilon$, and the bounded hypothesis. The coefficients of the third-order piezoelectric tensor $e_{ijk}^{h\varepsilon}$ (the coupled tensor), verifies the symmetry and bounded conditions. The second-order electric tensor $d_{ij}^{h\varepsilon}$ (dielectric permittivity), measured at constant strain, verifies the symmetry and bounded conditions (see [11,12]).

We recall that the variational problem (1)–(3) has a unique solution $(\mathbf{u}^{h\varepsilon}, \Phi^{h\varepsilon}) \in \mathbf{V}_{h\varepsilon}(\Omega_{h\varepsilon}) \times W_{h\varepsilon}(\Omega_{h\varepsilon})$, corresponding to *the saddle point* of this functional (see [11,12]):

$$
(\mathbf{v}, \Psi) \to \frac{1}{2} \int_{\Omega_{h\varepsilon}} \left( c^{h\varepsilon}(\mathbf{v}, \mathbf{v}) + 2e^{h\varepsilon}(\mathbf{u}, \Psi) - d^{h\varepsilon}(\Psi, \Psi) \right) dx - \int_{\Omega_{h\varepsilon}} \mathbf{f}\, \mathbf{v}\, dx,
$$

where

$$
\begin{cases}
c^{h\varepsilon}(\mathbf{u}, \mathbf{v}) & = & c_{ijkl}^{h\varepsilon}\, s_{ij}(\mathbf{u})\, s_{kl}(\mathbf{v}), \\
e^{h\varepsilon}(\mathbf{u}, \Psi) & = & e_{ikl}^{h\varepsilon}\, s_{kl}(\mathbf{u})\, \partial_i \Psi, \\
d^{h\varepsilon}(\Psi, \Psi) & = & d_{ij}^{h\varepsilon}\, \partial_i \Psi\, \partial_j \Psi,
\end{cases}
$$

and

$$
\begin{aligned}
\mathbf{V}_{h\varepsilon}(\Omega_{h\varepsilon}) & = \left\{ \mathbf{v} \in \mathbf{H}^1(\Omega_{h\varepsilon}),\ \mathbf{v} = \mathbf{0}\ \text{on}\ \partial\Omega_h \right\}, \\
W_{h\varepsilon}(\Omega_{h\varepsilon}) & = \left\{ \psi \in H^1(\Omega_{h\varepsilon}),\ \psi = 0\ \text{on}\ \partial\Omega_h \right\}.
\end{aligned}
$$

In order to study the different limit process ($h$ or $\varepsilon$ tends to zero).

## 3. The Thin Plate Behavior

We are interested in the limit of three-dimensional problem (1)–(3), when the thickness $h$ of the plate and the period $\varepsilon$ size of holes goes to zero.

### 3.1. Limit as the Thickness Tends to Zero

The first goal objective is to establish the limit of the three-dimensional variational problem associated with the problem (1) and (2), when $h \to 0$. We make a suitable choice of the orders of magnitude of the data (4) and (5), and using similar to those of Rahmoune [2] and Sène [4], in order to take into account the presence of holes, we can state the result below, which describes the limiting behavior of the electromechanical state when the thickness $h$ tends to zero.

**Theorem 1.** *For the piezoelectric variables* $(u_\alpha^{h\varepsilon}, u_3^{h\varepsilon}, \varphi^{h\varepsilon})$ *solution of three-dimensional problem associated the initial problem (1) and (2) defined in* $\Omega_{h\varepsilon}$. *We make the following assumptions about the magnitude of the data with respect to* $h$

$$
\begin{cases}
f_\alpha^h(x^{h\varepsilon}) = h^2 f_\alpha(x^\varepsilon),\ f_3^h(x^{h\varepsilon}) = h^3 f_3(x^\varepsilon) \\
g_\alpha^h(x^{h\varepsilon}) = h^3 g_\alpha(x^\varepsilon),\ g_3^h(x^{h\varepsilon}) = h^4 g_3(x^\varepsilon),
\end{cases}
,\ \alpha = 1, 2,\ \forall x^{h\varepsilon} \in \Omega_{h\varepsilon}
\tag{4}
$$

*where the couple $(f, g)$ is the element (independent of h) of $(L^2(\Omega_\varepsilon))^3 \times (L^2(\Omega_\varepsilon))^3$. Moreover, we assume that elastic, electric and piezoelectric constants are independent of h. We define the scaling of the unknowns:*

$$\begin{cases} u_\alpha^{h\varepsilon}(x^{h\varepsilon}) = h^2 u_\alpha^\varepsilon(h)(x^\varepsilon) \\ u_3^{h\varepsilon}(x^{h\varepsilon}) = h u_3^\varepsilon(h)(x^\varepsilon) \qquad \forall x^{h\varepsilon} \in \Omega_{h\varepsilon} \\ \varphi^{h\varepsilon}(x^{h\varepsilon}) = h^2 \varphi^\varepsilon(h)(x^\varepsilon) \end{cases} \tag{5}$$

*Then, when the thickness h tends to 0, we obtain*

$$\begin{cases} u_\alpha^\varepsilon(h) \rightharpoonup u_\alpha^{0\varepsilon}(x) = u_\alpha^{0\varepsilon}(x_1, x_2) - x_3^\varepsilon \partial_\alpha u_3^{0\varepsilon}(x_1, x_2), \\ u_3^\varepsilon(h) \rightharpoonup u_3^{0\varepsilon}(x_1, x_2), \\ \varphi^\varepsilon(h) \rightharpoonup \varphi^{0\varepsilon}(x_1, x_2). \end{cases} \tag{6}$$

*where the limits $u_\alpha^{0\varepsilon}(x_1, x_2)$, $u_3^{0\varepsilon}(x_1, x_2)$ and $\varphi^{0\varepsilon}(x_1, x_2)$ satisfy the solution of the variational problem*

$$\begin{cases} Find \quad (u_\alpha^{0\varepsilon}, u_3^{0\varepsilon}, \varphi^{0\varepsilon}) \in V_\varepsilon(\omega_\varepsilon) \times W_\varepsilon(\omega_\varepsilon) \times H^1(\omega_\varepsilon)/\mathbb{R}, \quad such\ as \\ \displaystyle\int_{\omega_\varepsilon} \left\{ N_{\alpha\beta}(u_\alpha^{0\varepsilon}, u_3^{0\varepsilon}, \varphi^{0\varepsilon}) s_{\alpha\beta}(v_\alpha^\varepsilon) + Q_\alpha(u_\alpha^{0\varepsilon}, \varphi^{0\varepsilon}) E_\alpha(\psi^\varepsilon) - M_{\alpha\beta}(u_3^{0\varepsilon}) \partial_{\alpha\beta} v_3^\varepsilon \right\} dx^\varepsilon \\ \quad = \displaystyle\int_{\omega_\varepsilon} \left\{ p_\alpha v_\alpha^\varepsilon + p_3 v_3^\varepsilon + m_\alpha \partial_\alpha v_3^\varepsilon \right\} dx^\varepsilon, \\ \forall \quad (v_\alpha^\varepsilon, v_3^\varepsilon, \psi^\varepsilon) \in V_\varepsilon(\omega_\varepsilon) \times W_\varepsilon(\omega_\varepsilon) \times H^1(\omega_\varepsilon)/\mathbb{R}, \end{cases} \tag{7}$$

*where*

$$V_\varepsilon(\omega_\varepsilon) = \left\{ v_\alpha \in H^1(\omega_\varepsilon), \ v_\alpha = 0 \ on \ \gamma_\varepsilon^{m0} \right\}$$

$$W_\varepsilon(\omega_\varepsilon) = \left\{ v_3 \in H^2(\omega_\varepsilon), \ v_3 = 0 \ and \ \partial_\nu v_3 = 0 \ on \ \gamma_\varepsilon^{m0} \right\},$$

$$m_\alpha = \int_{-1}^{+1} (x_3 f_\alpha + (g_\alpha^+ - g_\alpha^-)) dx_3, \quad p_\alpha = \int_{-1}^{+1} f_\alpha \, dx_3 \ and \quad p_3 = \int_{-1}^{+1} f_3 \, dx_3,$$

$$g^\pm = g|_{\Gamma_\varepsilon^\pm}, \ E_\alpha(\psi) = \frac{\partial\psi}{\partial x_\alpha},$$

$$\begin{cases} N_{\alpha\beta}(u_\alpha^{0\varepsilon}, u_3^{0\varepsilon}, \varphi^{0\varepsilon}) = \displaystyle\int_{-1}^{+1} \left\{ \hat{c}_{\alpha\beta\delta\tau}^\varepsilon \left[ s_{\delta\tau}(u_\alpha^{0\varepsilon}) - x_3 \partial_{\delta\tau} u_3^{0\varepsilon} \right] - \hat{e}_{\gamma\alpha\beta}^\varepsilon E_\gamma(\varphi^{0\varepsilon}) \right\} dx_3, \\ Q_\alpha(u_\alpha^{0\varepsilon}, \varphi^{0\varepsilon}) = \displaystyle\int_{-1}^{+1} \left\{ \hat{e}_{\gamma\alpha\beta}^\varepsilon s_{\alpha\beta}(u_\alpha^{0\varepsilon}) + \hat{d}_{\gamma\alpha}^\varepsilon E_\alpha(\varphi^{0\varepsilon}) \right\} dx_3, \\ M_{\alpha\beta}(u_3^{0\varepsilon}) = \displaystyle\int_{-1}^{+1} \left\{ -(x_3)^2 \hat{c}_{\alpha\beta\delta\tau}^\varepsilon \partial_{\delta\tau} u_3^{0\varepsilon} + x_3 \hat{c}_{\alpha\beta\delta\tau}^\varepsilon s_{\delta\tau}(u_\alpha^{0\varepsilon}) - x_3 \hat{e}_{\gamma\alpha\beta}^\varepsilon E_\gamma(\varphi^{0\varepsilon}) \right\} dx_3, \end{cases} \tag{8}$$

*where*

$$\begin{cases} \hat{c}_{\alpha\beta\delta\tau}^\varepsilon = c_{\alpha\beta\delta\tau}^\varepsilon - c_{3j\alpha\beta}^\varepsilon \tilde{b}_{3j3k}^\varepsilon \tilde{c}_{k3\alpha\beta}^\varepsilon + \dfrac{e_{3\alpha\beta}^\varepsilon \tilde{h}_{3\gamma\delta}^\varepsilon}{d_{33}^\varepsilon}, \\ \hat{e}_{\gamma\alpha\beta}^\varepsilon = e_{\gamma\alpha\beta}^\varepsilon - c_{3j\alpha\beta}^\varepsilon \tilde{b}_{3j3k}^\varepsilon \tilde{e}_{3k3}^\varepsilon + \dfrac{e_{3\alpha\beta}^\varepsilon \tilde{d}_{3\gamma}^\varepsilon}{d_{33}^\varepsilon}, \\ \hat{d}_{\gamma\alpha}^\varepsilon = d_{\gamma\alpha}^\varepsilon + e_{\gamma j3}^\varepsilon \tilde{b}_{3j3k}^\varepsilon \tilde{e}_{\gamma k3}^\varepsilon - \dfrac{d_{\alpha 3}^\varepsilon \tilde{d}_{\gamma 3}^\varepsilon}{d_{33}^\varepsilon}, \end{cases} \quad where \begin{cases} \tilde{c}_{k3\alpha\beta}^\varepsilon = c_{k3\alpha\beta}^\varepsilon + \dfrac{e_{3k3}^\varepsilon \tilde{e}_{3\alpha\beta}^\varepsilon}{d_{33}^\varepsilon}, \\ \tilde{e}_{\gamma k3}^\varepsilon = e_{\gamma k3}^\varepsilon + \dfrac{e_{3k3}^\varepsilon \tilde{d}_{\gamma 3}^\varepsilon}{d_{33}^\varepsilon}, \\ \tilde{h}_{3\alpha\beta}^\varepsilon = e_{3\alpha\beta}^\varepsilon - \dfrac{e_{3k3}^\varepsilon \tilde{b}_{k3j3}^\varepsilon e_{3\alpha\beta}^\varepsilon}{d_{33}^\varepsilon}, \\ \tilde{d}_{3\alpha}^\varepsilon = d_{3\alpha}^\varepsilon + e_{3k3}^\varepsilon \tilde{b}_{3k3j}^\varepsilon e_{\alpha j3}^\varepsilon, \end{cases} \tag{9}$$

*$\tilde{b}^\varepsilon$ is the inverse of $3 \times 3$ matrix $\left( c_{3j3k}^\varepsilon + \dfrac{e_{3j3}^\varepsilon e_{3k3}^\varepsilon}{d_{33}^\varepsilon} \right)_{3 \times 3}$.*

The proof of Theorem 1 is exactly the same as in Rahmoune [2,3] and in Sene [4,5]. We refer to Mechkour [12] for details.

### 3.2. Limit as the Period Tends to Zero

We now study the limit electrodynamic state when the period of perforation $\varepsilon$ tends to zero. Since the perforated plate has a periodic structure with period $\varepsilon$, this is a homogenization problem. We denote by $x$ the macroscopic variable and by $y = \frac{x}{\varepsilon}$ the microscopic variable. Let us define $\Omega_\varepsilon$ of periodically perforated subdomains of a bounded open set $\Omega$. The period of $\Omega_\varepsilon$ is $\varepsilon Y^*$, where $Y^*$ is a subset of the unit cube $Y = (0,1)^3$, which represented the solid domain. We use the two-scale convergence approach as introduced by Nguesteng [13] and Allaire [14]; we obtain

**Theorem 2.** *The sequences* $(u_\iota^\varepsilon)_{\varepsilon>0}$, $(\phi_\iota^\varepsilon)_{\varepsilon>0}$, $(\iota = 1,2)$ *two-scale convergences in* $u_\iota^{00}$, $\phi_\iota^{00}$, *respectively, where* $(u_\iota^{00}, \phi_\iota^{00})$ *is the unique solution of two-scale homogenized problem (Membrane plate equations)*

$$
\begin{cases}
-\mathbf{div}\ \boldsymbol{\text{œ}}^H(u_\iota^{00}, \phi^{00}) &=& \theta f & \text{in } \omega, \\[4pt]
-\mathbf{div}\ \mathbf{D}^H(u_\iota^{00}, \phi^{00}) &=& 0 & \text{in } \omega, \\[4pt]
u_\iota^{00} &=& 0 & \text{on } \partial\omega, \\[4pt]
\phi^{00} &=& 0 & \text{on } \partial\omega,
\end{cases}
\tag{10}
$$

*where* $\theta = \displaystyle\int_{Y^*} 1_{Y^*}(y)dy$ *represents the volume fraction on reference element. The new homogenized law is defined by*

$$
\begin{cases}
\sigma_{\alpha\beta}^H(u_\iota^{00}, \phi^{00}) &=& c_{\alpha\beta\zeta\eta}^H s_{\zeta\eta,x}(u_\iota^{00}) + e_{\zeta\alpha\beta}^H \dfrac{\partial\phi^{00}}{\partial x_\zeta}, \\[10pt]
D_\alpha^H(u_\iota^{00}, \phi^{00}) &=& -e_{\alpha\zeta\eta}^H s_{\zeta\eta,x}(u_\iota^{00}) + d_{\alpha\zeta}^H \dfrac{\partial\phi^{00}}{\partial x_\zeta}.
\end{cases}
\tag{11}
$$

*The homogenized coefficients* $c_{\alpha\beta\zeta\eta}^H$, $e_{\zeta\alpha\beta}^H$ *and* $d_{\alpha\zeta}^H$ *are defined by*

$$
c_{\alpha\beta\lambda\mu}^H = \left\langle \hat{c}_{\alpha\beta\zeta\eta}[\tau_{\lambda\mu}^{\zeta\eta} + s_{\zeta\eta,y}(w_\iota^{\lambda\mu})] + \hat{e}_{\zeta\alpha\beta}\dfrac{\partial q_\iota^{\lambda\mu}}{\partial y_\zeta}\right\rangle
\tag{12}
$$

$$
\begin{aligned}
e_{\delta\alpha\beta}^H &= \left\langle \hat{c}_{\alpha\beta\zeta\eta}s_{\zeta\eta,y}(\varphi^\delta) + \hat{e}_{\zeta\alpha\beta}[\delta_{\zeta\delta} + \dfrac{\partial\psi^\delta}{\partial y_\zeta}]\right\rangle \\[8pt]
&= \left\langle \hat{e}_{\delta\lambda\mu}[\tau_{\lambda\mu}^{\zeta\eta} + s_{\zeta\eta,y}(w_\iota^{\lambda\mu})] - \hat{d}_{\alpha\beta}\dfrac{\partial q_\iota^{\lambda\mu}}{\partial y_\beta}\right\rangle
\end{aligned}
\tag{13}
$$

$$
d_{\alpha\delta}^H = \left\langle -\hat{e}_{\alpha\zeta\eta}s_{\zeta\eta,y}(\varphi^\delta) + \hat{d}_{\alpha\beta}[\delta_{\beta\delta} + \dfrac{\partial\psi^\delta}{\partial y_\beta}]\right\rangle
\tag{14}
$$

*we use* $\langle . \rangle = \int_Y . \, dy$ *to denote the mean value over the basic cell* $Y$. *The local functions* $(w_\iota^{\zeta\eta}, q_\iota^{\zeta\eta})$ *and* $(\varphi^\delta, \psi^\delta)$ *are* $Y^*$*-periodic functions in* $y$, *independent of* $x$, *with solutions to these two local problems in* $Y^*$

$$
\begin{cases}
-\dfrac{\partial}{\partial y_\beta}\left\{\hat{c}_{\alpha\beta\zeta\eta}(x,y)[\tau_{\lambda\mu}^{\zeta\eta} + s_{\zeta\eta,y}(w_\iota^{\lambda\mu})] + \hat{e}_{\zeta\alpha\beta}(x,y)\dfrac{\partial q_\iota^{\lambda\mu}}{\partial y_\zeta}\right\} = 0 & \text{in } Y^*, \\[14pt]
-\dfrac{\partial}{\partial y_\alpha}\left\{-\hat{e}_{\alpha\zeta\eta}(x,y)[\tau_{\lambda\mu}^{\zeta\eta} + s_{\zeta\eta,y}(w_\iota^{\lambda\mu})] + \hat{d}_{\alpha\beta}(x,y)\dfrac{\partial q_\iota^{\lambda\beta}}{\partial y_\beta}\right\} = 0 & \text{in } Y^*, \\[14pt]
w_\iota^{\lambda\mu}, \ q_\iota^{\lambda\mu} \quad Y^*\text{-periodic,}
\end{cases}
\tag{15}
$$

*where*

$$\tau_{\lambda\mu}^{\zeta\eta} = \frac{1}{2}[\delta_{\zeta\lambda}\delta_{\eta\mu} + \delta_{\zeta\mu}\delta_{\eta\lambda}] \quad 1 \leq \lambda, \zeta, \mu, \eta \leq 2,$$

$$\begin{cases} -\dfrac{\partial}{\partial y_\beta}\Big\{\hat{c}_{\alpha\beta\zeta\eta}(x,y)s_{\zeta\eta,y}(\varphi^\delta) + \hat{e}_{\zeta\alpha\beta}(x,y)[\delta_{\zeta\delta} + \dfrac{\partial\psi^\delta}{\partial y_\zeta}]\Big\} = 0 \quad in\ Y^*, \\[4mm] -\dfrac{\partial}{\partial y_\alpha}\Big\{-\hat{e}_{\alpha\zeta\eta}(x,y)s_{\zeta\eta,y}(\varphi^\delta) + \hat{d}_{\alpha\beta}(x,y)[\delta_{\beta\delta} + \dfrac{\partial\psi^\delta}{\partial y_\beta}]\Big\} = 0 \quad in\ Y^*, \\[4mm] \varphi^\delta,\ \psi^\delta \quad Y^*\text{-}periodic. \end{cases} \tag{16}$$

*Furthermore, the sequence $(u_3^\varepsilon)_{\varepsilon>0}$ is the solution to problem (7) and (8). Two-scale convergence for $u_3^{00} \in H_0^2(\omega)$ shows that $u_3^{00}$ is unique solution for the two-scale homogenized problem (flexural plate equations):*

$$\begin{cases} \dfrac{\partial^2}{\partial x_\alpha \partial x_\beta}\Big(b_{\alpha\beta\gamma\tau}^H \dfrac{\partial^2}{\partial x_\gamma \partial x_\tau}(u_3^{00})\Big) &=& \theta(p_3 + \partial_\alpha m_\alpha) \quad &in\ \omega, \\[4mm] u_3^{00} &=& 0 &on\ \partial\omega, \\[4mm] \dfrac{\partial u_3^{00}}{\partial x_\nu} &=& 0 &on\ \partial\omega. \end{cases} \tag{17}$$

*The homogenized coefficient $b_{\alpha\beta\gamma\tau}^H$ is described by :*

$$b_{\alpha\beta\gamma\tau}^H = \langle \hat{c}_{\alpha\beta\zeta\nu}(x,y)\frac{\partial^2}{\partial y_\zeta \partial y_\nu}(\Pi_3^{\gamma\tau} + \chi_3^{\gamma\tau})\rangle, \tag{18}$$

*where $\Pi_3^{\gamma\tau}(y) = \frac{1}{2}y_\gamma y_\tau$. The local functions $\chi_3^{\gamma\tau}$ are defined by the solutions of cell problems*

$$\begin{cases} -\dfrac{\partial^2}{\partial y_\alpha \partial y_\beta}\Big\{\bar{\bar{c}}_{\alpha\beta\nu\zeta}(x,y)\dfrac{\partial^2}{\partial y_\nu \partial y_\zeta}(\chi_3^{\gamma\tau} + \Pi_3^{\gamma\tau})\Big\}\,dy = 0, \quad &on\ Y^* \\[4mm] \chi_3^{\gamma\tau} & Y^* - periodics, \\[4mm] \Pi_3^{\gamma\tau}(y) = \dfrac{1}{2}y_\gamma y_\tau. \end{cases} \tag{19}$$

The demonstration of Theorem 2 is exactly the same as in Mechkour [12]. We refer to Mechkour [11,12] for details.

The following result complements the two-scale convergence result by providing a strong convergence, which is very useful from a theoretical and numerical point of view. It is based on remarks that are admissible test functions (in the sense of Allaire [14]).

**Proposition 1.** *The following convergence holds when $\varepsilon$ goes to 0*

$$\begin{cases} 1_{\omega_\varepsilon}\Big(\dfrac{\partial^2 u_3^{0\varepsilon}}{\partial x_\alpha \partial x_\beta}(x) - \dfrac{\partial^2 u_3}{\partial x_\alpha \partial x_\beta}(x) - \dfrac{\partial^2 u_3^2}{\partial y_\alpha \partial y_\beta}(x,\frac{x}{\varepsilon})\Big) &\longrightarrow& 0 \quad in\ L^2(\omega)\ strongly, \\[4mm] 1_{\omega_\varepsilon}\Big(u_3^{0\varepsilon}(x) - u_3(x) - u_3^2(x,\frac{x}{\varepsilon})\Big) &\longrightarrow& 0 \quad in\ H^2(\omega)\ strongly. \end{cases}$$

## 4. Final Remarks

In this paper, we mathematically justify a reduced piezoelectric plate model, and we have rigorously established the limiting equations modeling the behavior of piezoelectric

plate in a periodically perforated domain, i.e., we have explicitly described forms of the homogenized coefficients of the elastic, dielectric and coupling tensors.

**Funding:** This research received no external funding.

**Conflicts of Interest:** The authors declares no conflict of interest.

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
