# Peer review of "Modeling of Perforated Piezoelectric Plates"

_mca, doi:10.3390/mca27060100_

Round 1

Reviewer 1 Report

The paper presents mathematical theory of homogenization for thin perforated piezoelectric plates. Technical work seems flawless. Significance of results and importance for various cases of interest (thin, thick, densely perforated plates) should be discussed. Consequences for modeling and numerics as well. Small typographical errors in text must be fixed.

Author Response

Dear Referee,

Thank you for your long and careful review!

Please find below our answers/comments to all your questions/remarks/suggestions.

We have modified and added the comments in the introduction and in the text.

We have corrected many mistake, and for all grammatical and typographical errors.

We added a new bibliography.

Thank you for your consideration!

Sincerely,

Dr. Mechkour

Reviewer 2 Report

In this paper, you note thata reduced piezoelectric plate model is mathemtically justified, and the limiting equations two modelling the behavior of piezoelectric plate is rigously established. but i think that the introduction doesn't provide sufficient background and include all relevant references. In addition, in the introduction, the purpose of this paper including creativity and originality is hardly expressed compared to the reference. In body section, you introduce some notations. Then, you recall the static three-dimensional piezoelectric model, of nonhomogeneous anisotropic thin plate, and you describe its formulation as a boundary value problem and the variational formulation. I think that this research design is a little appropriate. the results are not clearly presented and final remarks is only shortly described. Therefore, I think that quality of presentation and Interest to the readers is low. In order to publish this paper, it seems that you need to carefully check the entire paper.

Author Response

(The authors gave the same response as above.)

Round 2

Reviewer 1 Report

Satisfactory revision of the paper.

Reviewer 2 Report

The current form has been approved because all the requested modifications have been corrected.